# Enhanced Thermal Stability of Carbonyl Iron Nanocrystalline Microwave Absorbents by Pinning Grain Boundaries with SiBaFe Alloy Nanoparticles

**DOI:** 10.3390/nano14100869

**Published:** 2024-05-16

**Authors:** Yifan Xu, Zhihong Chen, Ziwen Fu, Yuchen Hu, Yunhao Luo, Wei Li, Jianguo Guan

**Affiliations:** 1School of Materials and Microelectronics, Wuhan University of Technology, Wuhan 430070, China; 2School of Science, Wuhan University of Technology, Wuhan 430070, China; 3State Key Laboratory of Advanced Technology for Materials Synthesis and Processing, Wuhan University of Technology, Wuhan 430070, China

**Keywords:** magnetic particles, grain size, thermal stability, pinning effect, microwave absorption

## Abstract

Nanocrystalline carbonyl iron (CI) particles are promising microwave absorbents at elevated temperature, whereas their excessive grain boundary energy leads to the growth of nanograins and a deterioration in permeability. In this work, we report a strategy to enhance the thermal stability of the grains and microwave absorption of CI particles by doping a SiBaFe alloy. Grain growth was effectively inhibited by the pinning effect of SiBaFe alloy nanoparticles at the grain boundaries. After heat treatment at 600 °C, the grain size of CI particles increased from ~10 nm to 85.1 nm, while that of CI/SiBaFe particles was only 32.0 nm; with the temperature rising to 700 °C, the grain size of CI particles sharply increased to 158.1 nm, while that of CI/SiBaFe particles was only 40.8 nm. Excellent stability in saturation magnetization and microwave absorption was also achieved in CI/SiBaFe particles. After heat treatment at 600 °C, the flaky CI/SiBaFe particles exhibited reflection loss below −10 dB over 7.01~10.11 GHz and a minimum of −14.92 dB when the thickness of their paraffin-based composite was 1.5 mm. We provided a low-cost and efficient kinetic strategy to stabilize the grain size in nanoscale and microwave absorption for nanocrystalline magnetic absorbents working at elevated temperature.

## 1. Introduction

With the progress in high-power and miniaturized electronic equipment, electromagnetic interference among heated components raises the urgent need for heat-resistant microwave-absorbing materials [1]. Additionally, heat-resistant microwave-absorbing materials are also required to control the radar cross-section (RCS) of hot parts of military devices [1,2,3]. At present, the widely used heat-resistant absorbents, such as silicon carbide [4], carbon-based composites [4,5], and barium titanate ceramics [6], exhibit excellent oxidation resistance and stable phases, while their microwave absorption mainly relies on dielectric loss, and there are difficulties in tuning permittivity dispersion over frequency [7,8], giving rise to narrow-band absorption and large thickness. In contrast, magnetic metallic absorbents, such as carbonyl iron (CI) [9], FeSiAl [10], FeCo [3], etc., exhibit both magnetic and dielectric loss to microwaves, in which the resonance and working band can be regulated by the composition and morphology [7]. When the grain size of the magnetic absorbents is on the nanoscale, strong inter-grain exchange coupling and consequent high permeability can be achieved [11,12]. Moreover, nanocrystalline magnetic absorbents are promising at elevated temperature due to their high Curie temperature (*T*_C_), e.g., *T*_C_ of Fe reaches around 770 °C, which gives rise to the potential to meet microwave absorption at elevated temperature with low thickness [13].

Nanocrystalline carbonyl iron (CI) is a soft magnetic absorbent with high initial permeability and low permittivity, thus exhibiting good reflection loss in the X (8~12 GHz) and Ku (12~18 GHz) bands [14]. However, when nanocrystalline CI powders are used at elevated temperature, they suffer from oxidation and grain instability. The oxidation caused by active chemical properties of component elements can be effectively inhibited by surface coating [10,15] and selected oxidation [16,17,18]. Nevertheless, the high-energy nanocrystalline grain boundaries can easily migrate and give rise to severe grain growth [19]. For example, the grains of Fe grew from 10 nm at room temperature to about 6 μm after heat treatment at 700 °C in a vacuum [20]. The coarsened grains affected the electromagnetic parameters via eddy current and ferromagnetic exchange. On one hand, a decrease in the amount of grain boundaries will enhance the electrical conductivity and eddy current [8], which leads to an increase in permittivity and decrease in permeability. On the other hand, according to the anisotropy model of random orientation, the initial permeability is inversely proportional to the sixth power of the grain size within the ferromagnetic exchange length [21,22]. Considering that the grain growth deteriorates the impedance matching of heat-resistant nanocrystalline magnetic absorbents, particular emphasis should be placed on stabilizing the grain size to achieve strong microwave absorption at elevated temperature [23].

From a kinetic perspective, curvature-driven migration of the grain boundaries accounts for the grain growth [24]. Chemically ordered crystal structures, such as the Fe_3_Si superlattice [25], can inhibit atomic diffusion and, thus, hinder the migration of grain boundaries [26], while their formation in Fe-based alloys is limited to certain elements. The pinning effect has been widely used to stabilize the grains of nanocrystalline particles, such as solute/second-phase dragging [27,28,29]. The thermal stability of nanocrystalline Fe/10 wt.% Al alloy was investigated as a typical example, which showed that the generation of the AlN and Al_2_O_3_ phases effectively hindered grain growth from room temperature to 1200 °C [30]. However, these pinning phases, such as oxides [26,29] and nitrides [30], were often coarsened at high temperature and led to the weakened stabilization of grains [28]. Metallic solutes, such as Zr [31] and Cr [27,31], exhibited lower diffusion than Fe and suppressed grain growth, while they easily formed a solid solution with a matrix at elevated temperature, resulting in rapid grain growth. Therefore, an ideal pinning phase ought to have low solid solubility, a coherent interface, and low diffusion rate in the nanocrystalline CI matrix [26].

In this work, we propose a kinetic strategy to stabilize nanograins in CI particles via the pinning effect. In our strategy, the pinning effect of the SiBaFe alloy was introduced to CI particles by ball milling. The significant difference in the atomic radius among Si, Ba, and Fe prevented the formation of a solid solution; meanwhile, Fe in the SiBaFe alloy helped enhance wettability with the Fe matrix and, thus, ensured the thermal stability of Si and Ba. The obtained CI/SiBaFe composite particles exhibited excellent grain thermal stability, the average grain size of which retained 32.0 nm and 40.9 nm instead of 85.1 nm and 158.1 nm in pure CI after annealing at 600 °C and 700 °C, respectively. Consequently, enhanced saturation magnetization, impedance matching, and microwave absorption were achieved at elevated temperature.

## 2. Materials and Methods

CI particles and SiBaFe alloy powders were purchased from Shaanxi XingHua Chemistry Share Co., Ltd. (Xingping, China) and Anyang GuangSheng Resistant Metallic Material Co., Ltd. (Anyang, China), respectively. Cyclohexane of analytical purity was purchased from Sinopharm Chemical Reagent Co., Ltd. (Shanghai, China), without further purification.

Flaky CI/SiBaFe particles were obtained by a two-step method of wet ball milling and attritor ball milling. Raw CI and SiBaFe powders to a total of 20 g were first added to a milling tank with 400 g of 316L stainless-steel balls and 20 mL of cyclohexane as the process control agent. The mixture was then milled for 80 h, 120 h, and 160 h at 300 r/min. The CI/SiBaFe powders were collected by a filter followed by drying at 60 °C for 3 h. The CI/SiBaFe powders were then milled in an attritor with 800 g of ZrO_2_ balls and 100 mL of cyclohexane for 8 h at rotation rate of 150 r/min. The resultant flaky CI/SiBaFe particles were collected and subsequently dried at 60 °C for 3 h. In this study, different contents of SiBaFe alloy particles were used to prepare CI/SiBaFe particles, i.e., 5 wt.%, 10 wt.%, and 15 wt.%; the resultant samples were then denoted as CI/SiBaFe-5, CI/SiBaFe-10, and CI/SiBaFe-15, respectively. To test the heat resistance of obtained CI/SiBaFe particles, the powders were annealed at 300~700 °C for 1 h in vacuum.

The morphology of CI/SiBaFe powders was investigated by scanning electron microscopy (SEM, Hitachi S-4800, Hitachi, Japan). The compositions were confirmed by energy-dispersive spectroscopy (EDS) attached to SEM. The crystal structure, grain size, and internal strain of CI/SiBaFe particles were characterized by X-ray diffraction (XRD: Cu K*α*-1.54060 Å, Bruker D8 Advance, Ettlingen, Germany). Vibrating sample magnetometer (VSM, LakeShore 7404S, Carson, CA, USA) was used to measure the static magnetic properties of powders. The oxidation kinetics of flaky CI and CI/SiBaFe particles was investigated by integrated thermal analysis (Netzsch STA449F3, Hanau, Germany). The samples were heated from room temperature to 1000 °C in air at a rate of 10 °C/min. The thermogravimetric (TG) and differential scanning calorimetric (DSC) curves were recorded during the heating process. To characterize the relative complex permittivity (*ε*_r_) and permeability (*μ*_r_) of CI/SiBaFe particles, powders were homogeneously dispersed in paraffin with a mass ratio of 3.265:1, and the mixture was then pressed into a coaxial ring shape, with outer and inner diameters of 7 and 3 mm and heights of 2~3 mm, respectively. The electromagnetic parameters *ε′*, *ε″*, *μ′* and *μ″* of coaxial samples were measured using vector network analyzer (VNA, Keysight N5230A, Santa Rosa, CA, USA).

## 3. Results and Discussion

Flaky CI/SiBaFe particles were obtained through a two-step method based on wet and attritor ball milling. Figure 1a illustrates the preparation of CI/SiBaFe particles by wet ball milling. As shown in Figure 1b,c, raw CI particles were in a spherical shape with a diameter less than 5 μm, while raw SiBaFe alloy particles exhibited a large distribution in dimensions and showed angular fragments, with an average diameter much larger than that of CI. The input of milling energy caused the alloying of CI with SiBaFe, and the preliminarily prepared CI/SiBaFe particles exhibited irregular powders instead of spherical or fragmentary (see Figure 1d and Appendix A), accompanied by the refinement of grains (see Appendix A). The effect of SiBaFe contents on the alloying was investigated by XRD, as shown in Figure 1e–g. The raw CI particles mainly consisted of the *α*-Fe phase (see Figure 1e), while the raw SiBaFe alloy was mainly composed of elemental Si and compounds such as FeSi_2_, BaSi_2_ and Ba_2_Si (see Figure 1f and Appendix A). All CI/SiBaFe particles with varying SiBaFe contents mainly consisted of the *α*-Fe phase after milling for 160 h (see Figure 1g). No evident diffraction peak of the raw SiBaFe alloy was observed, revealing that the SiBaFe particles were ground to powders and successfully alloyed with CI particles by ball milling. The difference in the atomic radius among Fe (1.26 Å), Si (1.17 Å), and Ba (2.24 Å) prevented the formation of a solid solution, since no evident shift in diffraction peaks was observed. Also, no diffraction peak of oxides was observed due to the isolation of oxygen from the powders by the sealed milling tank. The EDS images (see Figure 1h–j and Appendix A) indicate that the Fe, Si, and Ba elements were uniformly distributed in CI/SiBaFe particles, without evident segregation.

To enhance the initial magnetic permeability, shape anisotropy was introduced to differentiate the out-of-plane and in-plane magnetic anisotropy fields. Subsequent attritor ball milling was thus employed to obtain flaky CI/SiBaFe particles. As shown in Table 1, the detected contents catered to the feeding ratio. For instance, Fe, Si, and Ba occupied 86.53 wt.%, 8.34 wt.%, and 5.13 wt.% (except C and O), respectively, in flaky CI/SiBaFe-15 particles.

To investigate the effect of SiBaFe content on the heat resistance, CI, CI/SiBaFe-5, CI/SiBaFe-10, and CI/SiBaFe-15 were annealed at 300~700 °C for 1 h in a vacuum. The XRD spectra of particles before and after annealing are shown in Figure 2a–d, respectively. The main phase in flaky CI/SiBaFe particles with different SiBaFe contents maintained *α*-Fe after heat treatment. However, the diffraction peaks exhibited a significant difference in the full width at half maximum (FWHM), indicating variation in the dimensions of grains. The FWHM of the *α*-Fe diffraction peak was positively correlated with SiBaFe contents, which indicated that SiBaFe doping optimized the grain stability of flaky CI/SiBaFe particles. To quantify the thermal stability of grains in particles, the corresponding grain size (*D*) and internal strain (*ε*) of CI/SiBaFe particles were calculated by the Williamson–Hall model [32]:(1)$$\beta \cos\theta =4\sin\theta \cdot \varepsilon +\frac{K\lambda }{D}$$
where *β* is the FWHM, *θ* is the angle of diffraction peak, *λ* is the incident wavelength of X-ray equal to 1.54060 Å, and constant *K* equals 0.89. The results are shown in Figure 2e,f. It can be noticed that the average grain sizes of flaky CI/SiBaFe particles were all about 9 nm at room temperature and did not show an obvious difference after annealing at 300~400 °C, while the growth rate accelerated significantly when the annealing temperature exceeded 500 °C. After annealing at 700 °C, the grain size of CI particles reached 158.10 nm, which increased by 16.8-times compared with 9.38 nm at room temperature. In contrast, the grain size of CI/SiBaFe-15 increased to 40.9 nm, while it was significantly smaller than that of CI. In addition, the internal strain of flaky CI/SiBaFe particles (CI/SiBaFe-5, CI/SiBaFe-10, and CI/SiBaFe-15) decreased with an increase in the annealing temperature, and the decline rate was slightly faster than that of CI particles. A slight increase in *ε* was observed in CI/SiBaFe-15 when the heat treatment exceeded 500 °C, which may have been caused by the different growth rate and local distortion between pinning particles and matrix grains [33]. The variation in grain size and internal strain indicated that the grain growth was effectively hindered by SiBaFe doping.

The superlattice structure in the FeSi alloy after heat treatment has been widely reported [34,35,36]. However, no superlattice diffraction peak was observed in the XRD spectra of flaky CI/SiBaFe particles. This may have been due to the low SiBaFe contents and the preparation method of ball milling, in which Si and Ba could not easily form a solute with the Fe matrix at the atomic level. No evident shift in the diffraction peaks was observed in the XRD spectra of CI/SiBaFe particles, indicating the absence of a solid solution. Instead, most Si and Ba elements existed at the grain boundaries and did not further dissolve into the *α*-Fe lattice. From a mechanical perspective, grain boundaries stop migrating when the interfacial tensions from every direction achieve a balance. Pressure towards the center of curvature at the interface thus generates, which acts positively on the convex grain and negatively on the concave grain. The pressure difference gives rise to the difference in free energy (∆*G*) [24]:(2)$$\Delta G=V_{m}\Delta p=V_{m}\gamma \left(\frac{1}{\rho_{1}}+\frac{1}{\rho_{2}}\right)$$
where *V*_m_ is the molar volume, *γ* is the surface tension coefficient, and *ρ*_1_ and *ρ*_2_ are the radius of curvature in the convex and concave grains, respectively. The ∆*G* drives the atoms to transition from the convex to the concave grains, indicating that the migrating tendency of grain boundaries is negatively correlated with the radius of curvature (*ρ*). Research has revealed that the pinning effect provided by the solutes or second phases inhibits the grain growth by enlarging *ρ* [26,29,30,37]. Therefore, SiBaFe particles at grain boundaries hindered the displacement of grain boundaries. The optimized thermal stability of grains in flaky CI/SiBaFe particles was closely related to the pinning effect of SiBaFe. Furthermore, previous research indicated that the coherent interface with a matrix, in turn, enhanced the thermal stability of the pinning phases, which required favorable wettability between the two [26,28,30,31]. Otherwise, the grain growth in pinning phases at elevated temperature would severely deteriorate the pinning effect and thermal stability of matrix grains [28]. In flaky CI/SiBaFe particles, Fe from the SiBaFe alloy exhibited excellent wettability with carbonyl iron, which ensured the close integration between the pinning phases and matrix. Therefore, flaky CI/SiBaFe particles exhibited enhanced thermal stability of grains, even after heat treatment at 700 °C.

In addition to grain stability, an anti-oxidation phenomenon was also found after doping SiBaFe in CI. Flaky CI/SiBaFe particles were heated from room temperature to 1000 °C in air, at a rate of 10 °C/min. As shown in Figure 3, the oxidation kinetics of particles was investigated by TG and DSC curves. The results revealed that, with an increase in the SiBaFe contents, the temperature of the weight gain caused by oxidation in flaky CI/SiBaFe particles also gradually increased. For CI particles, their mass reached a maximum of 134.4% at about 450 °C and then barely increased. In a study by Yin et al., flaky CI particles gained a mass of 138.43% after heat treatment in air, which was close to that of Fe_3_O_4_ in theory [38]. The slightly lower weight gain in our work may be caused by minor impurities or partial oxidation on the surface. Compared with CI, at 450 °C, CI/SiBaFe-5, CI/SiBaFe-10, and CI/SiBaFe-15 exhibited a lower weight gain of 109.30%, 105.43%, and 104.22%, respectively. Although the masses of the samples all increased by about 34% at 1000 °C, the TG curves exhibited a significant difference in the weight-gain tendency. As shown in Figure 3b, the DSC curves revealed that, with an increase in SiBaFe contents, the first endothermic peak gradually shifted from ~401 °C to ~531 °C. In flaky CI/SiBaFe-15 particles, another endothermic peak appeared at ~904 °C. This was because the Si and Ba atoms preferentially bonded with O atoms, since their oxidative activity is superior to Fe [39,40]. At elevated temperature, oxide layers formed due to the oxidation of Si and Ba, which protected CI from interaction with oxygen [15,18]. Inhibition of O diffusion, thus, slowed down the oxidation, and the second endothermic peaks in CI/SiBaFe-10 and CI/SiBaFe-15 may have been caused by the oxidation of Fe after Si drained.

The effect of SiBaFe doping on the static magnetic properties of flaky CI/SiBaFe particles is shown in Figure 4 and Appendix A. In Figure 4a, except for the slight increase at temperatures below 300 °C, the saturation magnetization (*M*_S_) of CI exhibited a significant drop after heat treatment and only retained 71.49 emu/g when the temperature reached 700 °C. The internal strain induced by milling was first released, while the severe grain growth during heat treatment led to a decrease in the amount of grains with a ferromagnetic exchange length and, thus, weakened the ferromagnetic exchange coupling and the consistency of atomic magnetic moments during magnetization [41]. Consequently, the *M*_S_ of CI decreased at elevated temperature. In contrast, as shown in Figure 4b,c, due to the pinning effect of SiBa, the *M*_S_ of flaky CI/SiBaFe particles remained basically stable throughout heat treatment and exceeded that of pure CI particles at 500~700 °C. Compared with CI, doping of non-ferromagnetic SiBaFe led to a gradual decrease in *M*_S_ with an increase in the doping content; however, the *M*_S_ for SiBaFe particles were stable due to the enhanced thermal stability of grain.

Figure 5 exhibits the evolution of the relative complex permittivity (*ε*_r_ = *ε′* − *iε″*) and complex permeability (*μ*_r_ = *μ′* − *iμ″*) of CI and CI/SiBaFe-15 after heat treatment at different temperatures. As shown in Figure 5a,b, at 1 GHz, the *ε′* of CI particles increased from 21.09 at room temperature to 173.60 at 700 °C, and *ε″* increased from 2.80 at room temperature to 73.45 at 700 °C. The significant increase in complex permittivity was caused by increased conductivity and enhanced interfacial polarization [8,42]. The relationship between the grain size and volume fraction of grain boundaries (*f*_GB_) can be described as [43]:(3)$$f_{\mathrm{GB}}=1-\frac{{\left(D-\bar{d}\right)}^{3}}{D^{3}}$$
where $\bar{d}$ represents the effective thickness of the grain boundary, which is mainly affected by the atomic radius of the alloying element. Since CI contained no other alloying elements, the value of $\bar{d}$ was stable throughout heat treatment. Therefore, grain growth and the consequent decrease in grain boundaries in unit volume reduced electron scattering, resulting in an increase in conductivity and complex permittivity. The accelerated increasing rate of complex permittivity coincided with obvious grain growth when the annealing temperature reached 500 °C. In contrast, the relative complex permittivity of CI/SiBaFe-15 before and after annealing is shown in Figure 5e–f. On one hand, SiBaFe doping reduced the plasticity of composite particles due to the fragility of SiBaFe. The obtained flaky CI/SiBaFe particles exhibited a small average diameter, which reduced the possibility for the formation of a conductive network among particles. On the other hand, low conductivity of the doped Si and Ba led to the weakened electron transport capacity of flaky CI/SiBaFe-15 particles [44,45,46]. Consequently, the *ε′* and *ε″* of CI/SiBaFe-15 were significantly lower than that of CI at room temperature. From room temperature to 400 °C, the growth of grains and decrease in defects weakened electron scattering in CI/SiBaFe-15 and accounted for the slight increase in complex permittivity. However, when the annealing temperature exceeded 400 °C, the complex permittivity decreased. The TG and DSC curves shown in Figure 3 indicate the oxidation of Si and Ba. In our heat treatment, air may have reacted with CI/SiBaFe due to the low vacuum. The oxides at grain boundaries weakened the eddy current and exhibited low conductivity [15,18], resulting in a decrease in complex permittivity of over 400 °C.

As for the relative complex permeability, the results revealed that the real and imaginary parts of the complex permeability of CI both gradually decreased at elevated annealing temperature (see Figure 5c,d). At 1 GHz, the *μ′* of CI decreased from 3.57 at room temperature to 2.28 at 700 °C, and the maximum of *μ″* decreased from 1.54 at room temperature to 0.51 at 700 °C. Considering the exchange coupling among grains, the initial permeability (*μ*_i_) was affected by *M*_S_ and *D* [21,22]:(4a)$$D<L_{\mathrm{ex}},\;\mu_{i}=p_{u}\frac{M_{S}^{2}}{\mu_{0}<K>}\approx p_{u}\frac{M_{S}^{2}\cdot A^{3}}{\mu_{0}K_{1}^{4}\cdot D^{6}}.$$
(4b)$$D=L_{\mathrm{ex}},\;\mu_{i}=p_{u}\frac{M_{S}^{2}}{\mu_{0}K_{1}}$$
(4c)$$D>L_{\mathrm{ex}},\;\mu_{i}=p_{u}\frac{M_{S}^{2}\cdot D}{\mu_{0}\sqrt{AK_{1}}}$$
where *L*_ex_ is the ferromagnetic exchange length equal to about 35 nm in Fe-based nanocrystalline [47], <*K*> is the effective anisotropy coefficient, which is proportional to the sixth power of *D*, *K*_1_ is the magnetocrystalline anisotropy coefficient, *μ*_0_ is permeability in a vacuum, *A* is the exchange coefficient, and *p*_u_ is a constant. Equation (4) indicates that the permeability first rapidly decreased with grain growth within a critical grain size and then changed significantly. According to Equation (4a), the average grain size of CI was retained within the exchange length after heat treatment at 400 °C (~15.8 nm), giving rise to the enlarged magnetocrystalline anisotropy and the rapid decrease in permeability. When the annealing temperature exceeded 500 °C, the average grain size of CI increased to ~46.2 nm, and the *μ″* first increased according to the proportional relationship between *μ*_i_ and *D* in Equation (4c), while experiencing a subsequent decrease due to the variation in *M*_S_ and *K*_1_. During the whole heat treatment, *M*_S_, the quadratic term, which decreased at elevated temperature due to the grain growth and consequently weakened the ferromagnetic exchange coupling, would lead to a significant decrease in complex permeability.

The relative complex permeability of CI/SiBaFe-15 is shown in Figure 5g,h. At 1 GHz, *μ′* decreased from 3.22 at room temperature to 2.52 at 700 °C, and the peak value of *μ″* decreased from 1.38 at room temperature to 0.93 at 700 °C, which indicated that both decrements in the *μ′* and *μ″* of CI/SiBaFe-15 were much smaller than that of CI particles due to the stable grain size. The results revealed that inhibited grain growth by SiBaFe doping helped stabilize the electromagnetic parameters at elevated temperature. The average grain size of CI/SiBaFe-15 was retained far below the exchange length, even after annealing at 500 °C (~19.5 nm), thus exhibiting relatively stable complex permeability. When the annealing temperature exceeded 600 °C, the average grain size gradually increased and reached ~40.9 nm after heat treatment at 600 °C. The consequently weakened ferromagnetic exchange coupling gave rise to a decrease in *μ″*.

The reflection loss (*RL*) and the impedance matching performance of flaky CI and CI/SiBaFe particles were calculated. According to the transmission line theory, the *RL*, reflectivity from CI/SiBaFe paraffin-based composites, can be calculated by [48]:(5a)$$RL=-20\mathrm{lg}\left|\frac{Z_{\mathrm{in}}-Z_{0}}{Z_{\mathrm{in}}+Z_{0}}\right|$$
(5b)$$Z_{in}=Z_{0}\sqrt{\frac{\mu_{r}}{\varepsilon_{r}}}tanh(j(\frac{2\pi ft}{c})\sqrt{\mu_{r}\varepsilon_{r}})$$
where *Z*_0_ is the air impedance equal to 377 Ω, *Z*_in_ and *t* are the input impedance and thickness of the absorbing material, respectively, and *c* is the speed of light. The calculated *RL* when *t* = 1.5 mm is shown in Figure 6a,b. For CI particles, the absorption peak shifted to a lower frequency when heated at elevated temperature, with significant deterioration of the *RL*, the minimum of which decreased from −13.13 dB at room temperature to −8.60 dB at 600 °C and further to −2.70 dB at 700 °C. In Figure 6c, it can be seen that the increased complex permittivity of CI caused by the growth of grains and removal of defects worsened the impedance matching, resulting in deteriorated microwave absorption at high temperature. For flaky CI/SiBaFe-15 particles, although they exhibited weaker absorption than CI at room temperature, the *RL* retained stable after heat treatment and was even enhanced at 600~700 °C, which was caused by thermally stable permittivity and magnetic loss. CI/SiBaFe-15 exhibited effective absorption (<−10 dB), ranging from 7.04 to 10.11 GHz at 600 °C, with an increase in the *RL* minimum from −10.49 dB at room temperature to −14.92 dB. As can be seen in Figure 6d, due to the doping of non-magnetic Si and Ba, CI/SiBaFe-15 exhibited lower permeability and consequent impedance matching. Nevertheless, the grain growth inhibited by the SiBaFe pinning effect cooperated with slight oxidation to inhibit the increase in permittivity when the annealing temperature exceeded 500 °C, thus optimizing the impedance matching. From the calculated results, SiBaFe doping was expected to enhance the microwave absorption of flaky CI/SiBaFe particles at 500~700 °C.

## 4. Conclusions

In summary, enhanced microwave absorption at elevated temperature was achieved for flaky CI/SiBaFe particles by inhibiting the grain growth. The SiBaFe alloy was alloyed with carbonyl iron particles via a two-step ball milling method. The pinning phases at the grain boundaries proved to be thermally stable and effectively inhibited the migration of grain boundaries at elevated temperature. The grain size remained at 32.0 nm after annealing at 600 °C and 40.8 nm at 700 °C. The *M*_S_ also remained stable after annealing at 300~700 °C due to the ferromagnetic exchange coupling among pinned nanocrystalline grains. After doping at 15 wt.% SiBa, when heated from room temperature to 700 °C, the increment in *ε′* decreased considerably from 152.51 to nearly 0 at 1 GHz, and the decrement in the *μ″* maximum decreased from 1.03 to 0.45. The simulated *RL* of the resultant absorbents in paraffin reached a minimum of −14.92 dB at 1.5 mm and exhibited effective absorption (<−10 dB) with 7.01~10.11 GHz at 600 °C. This work provided an effective route to inhibit grain growth at elevated temperature, and the obtained flaky CI/SiBaFe particles showed great potential for heat-resistant absorbents.

## Figures and Tables

**Figure 1 nanomaterials-14-00869-f001:**
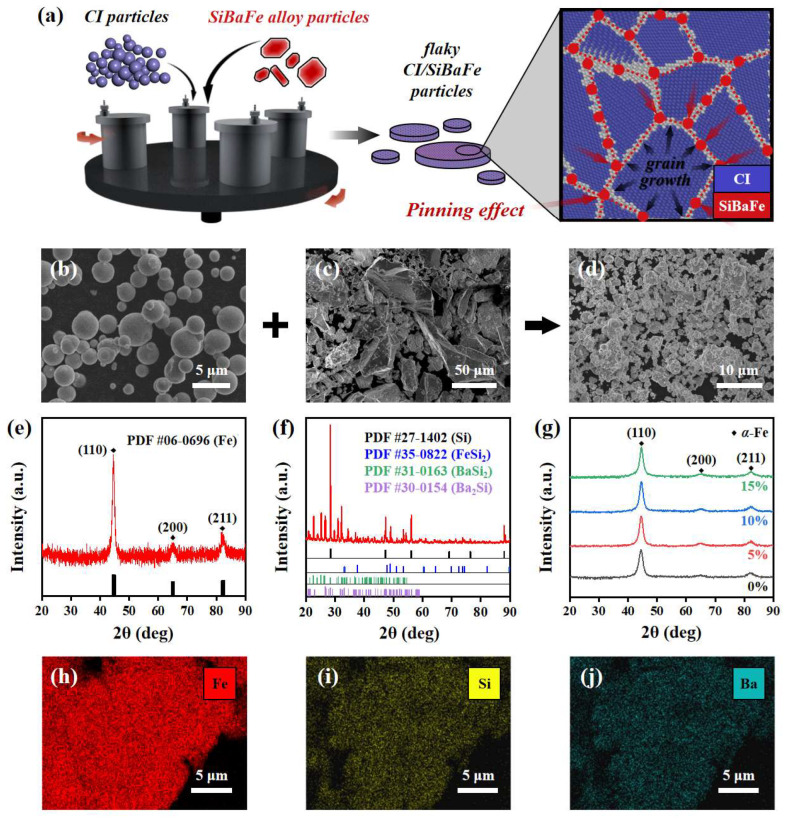
Preparation of CI/SiBaFe particles by wet ball milling: (**a**) sketch for alloying of CI and SiBaFe particles; (**b**–**d**) SEM images of (**b**) raw CI particles, (**c**) raw SiBaFe particles, and (**d**) CI/SiBaFe-15 composite particles; (**e**–**g**) XRD spectra of (**e**) raw CI particles, (**f**) raw SiBaFe alloy particles, and (**g**) CI/SiBaFe composite particles with different SiBaFe contents; (**h**–**j**) distribution of (**h**) Fe, (**i**) Si, and (**j**) Ba in CI/SiBaFe-15 composite particles.

**Figure 2 nanomaterials-14-00869-f002:**
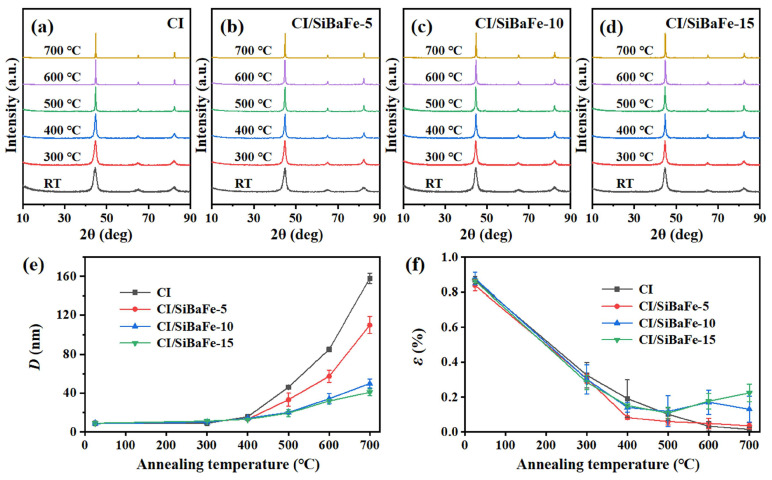
Thermal stability of grains in flaky CI/SiBaFe particles after heat treatment at different temperatures: (**a**–**d**) XRD spectra for (**a**) CI, (**b**) CI/SiBaFe-5, (**c**) CI/SiBaFe-10, and (**d**) CI/SiBaFe-15 particles; (**e**) grain size; and (**f**) internal strain.

**Figure 3 nanomaterials-14-00869-f003:**
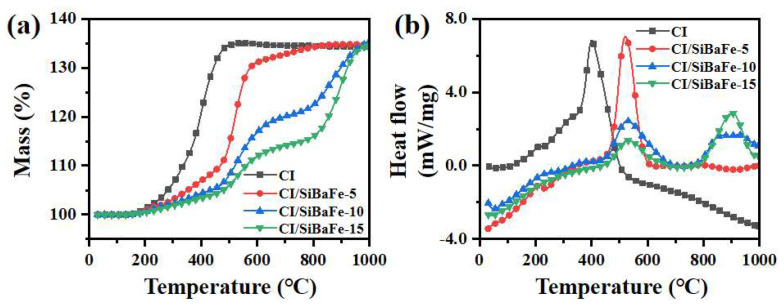
Oxidation kinetics of flaky CI/SiBaFe particles: (**a**) TG and (**b**) DSC curves of CI, CI/SiBaFe-5, CI/SiBaFe-10 and CI/SiBaFe-15.

**Figure 4 nanomaterials-14-00869-f004:**
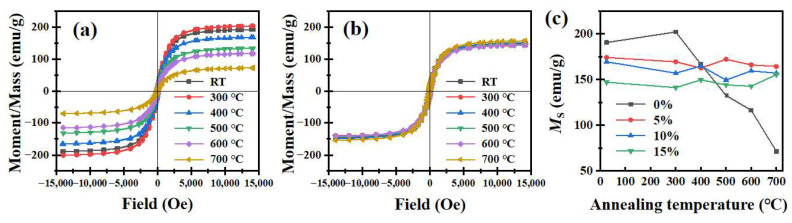
Thermal stability of static magnetic properties of flaky CI/SiBaFe particles: hysteresis loops of (**a**) CI and (**b**) CI/SiBaFe-15 before and after heat treatment at different temperature in vacuum; (**c**) evolution of *M*_S_ of flaky CI and CI/SiBaFe particles at different temperature.

**Figure 5 nanomaterials-14-00869-f005:**
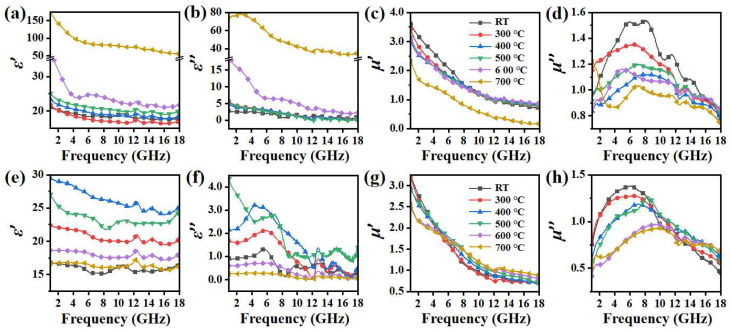
Effects of SiBaFe doping on the thermal stability of electromagnetic parameters: relative complex permittivity and permeability of (**a**–**d**) flaky CI and (**e**–**h**) CI/SiBaFe-15 particles before and after heat treatment.

**Figure 6 nanomaterials-14-00869-f006:**
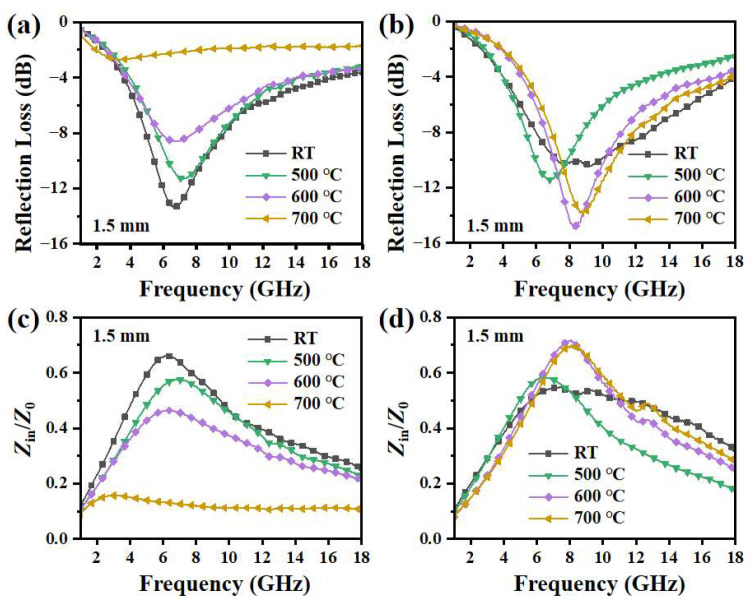
(**a**,**b**) Microwave absorption of (**a**) CI and (**b**) CI/SiBaFe-15 particles; (**c**,**d**) impedance matching analysis for (**c**) CI and (**d**) CI/SiBaFe-15 particles.

**Table 1 nanomaterials-14-00869-t001:** Composition of flaky CI/SiBaFe composite particles.

Sample	Fe (wt.%)	Si (wt.%)	Ba (wt.%)
CI	~100	—	—
CI/SiBaFe-5	94.80 ± 4.95	3.05 ± 1.04	2.15 ± 1.36
CI/SiBaFe-10	90.68 ± 5.18	5.56 ± 1.14	3.76 ± 1.34
CI/SiBaFe-15	86.53 ± 4.67	8.34 ± 1.26	5.13 ± 2.79

## Data Availability

The data presented in this study are available on request from the corresponding author.

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
