# Peer review of "Enhanced Thermal Stability of Carbonyl Iron Nanocrystalline Microwave Absorbents by Pinning Grain Boundaries with SiBaFe Alloy Nanoparticles"

_nanomaterials, 2024, doi:10.3390/nano14100869_

Round 1

Reviewer 1 Report

Comments and Suggestions for Authors

     The manuscript “nanomaterials-2942926” describes a method for increasing the thermal stability of a microwave radiation absorber based on nanocrystalline carbonyl iron particles. Increased thermal stability was achieved by doping the particles with SiBaFe alloy. The article is interesting and can be published after revision.

1.      In the Title: "... kinetically pinning of grain boundaries..." wouldn't it be better to replace it with "pinning of grain boundaries", since the term "kinetically pinning" is never used throughout the entire article?

2.      The Abstract should be edited to be written more clearly. For example, correct the following unclear statement: “After heat treatment at 600 ℃, the flaky CI/SiBaFe particles exhibited reflection loss below -10 dB 22 over 7.01~10.11 GHz and minimum of -14.92 dB at 1.5 mm”. The Abstract does not explain what 1.5 mm means. The abstract should also clarify that micron-sized particles with nanometer-sized grains were studied.

3.      Advantages of the material developed in this work should be discussed compared to similar microwave absorbents in more detail.

4.      The equations need to be checked and corrected.

-          For example, in equation (2), γ is incorrectly referred to as interfacial pressure (see line 195). However, γ is the surface tension coefficient.

-          In line 195, the confusing term “molar volume of grain boundaries” should be corrected. The term "molar volume" refers to the volume of one mole of a substance, not a geometric object.

-          Equation (2) is immediately followed by equation (4), that is, equation (3) is skipped.

5.      It is necessary to explain the choice of SiBaFe alloy for the most effective pinning of grain boundaries.

6.      All parameters used in the work should be collected in the “Nomenclature” section.

7.      In the “Conclusions” section, the main advantage of the material developed in the work should be clearly indicated.

Author Response

We really appreciate the positive comment of the reviewer for our manuscript, which we believe can encourage us for future work. The revised parts in the manuscript have been highlighted in red. Details can be seen in the attachment.

Reviewer 2 Report

Comments and Suggestions for Authors

The topic of the presented manuscript is of interest due to the need to increase the thermal stability of materials absorbing microwave radiation. In general, the presentation of the results is quite well organized, however, meanwhile, there are some comments on the manuscript.

1. In Fig.1b, it is impossible to see evidence of the flat shape of the particles: from the SEM photos, you can tell that they are just spherical. And in Fig. i and especially j find it very difficult to see anything.

2. Section 2 indicates that the equipment used was “XRD: Cu Ka-1.54060 Å , Bruker D8 Advance, German).” However, later in section 3, a different wavelength of X-ray radiation was used to calculate the size of the crystallites.

3. Samples with "coaxial ring shape with outer and inner diameters of 7 and 3 mm, respectively" were used for microwave measurements. I would like to know the height of these rings in order to compare them with the possibility of making correct measurements at the applied frequencies.

4. How correct is it to use expression (4) in numerical estimates for flaky particles, which have a large aspect ratio, curvature and a fairly noticeable size spread? 

5. Was the type of particle placement in the matrix controlled during the manufacture of the composite? After all, with such a particle shape, it is possible to form a noticeable texture and pronounced anisotropy of the properties of the final material?

6. «The significant increase of complex permittivity was caused by increased conductivity and enhanced interfacial polarization». This is quite strange, since at high temperatures almost all iron must be oxidized to magnetite, which is not more conductive than carbonyl iron.

7. How justified is the use of a model using the idea of ferromagnetic exchange length (expressions 5-1, 5-2, 5-3), with such a pronounced anisotropy of the particle shape? Have the authors tried to introduce a correction taking into account the magnetostatic interaction in the system?

8. Line 318: it seems worth clarifying that the expressions given to describe the behavior of the material refer to the condition of reflection from the metal surface.

9. The general conclusion regarding the thermal stability of the material should provide for the repeatability of the results when repeating the experiment with a cyclic temperature change.

Author Response

(The authors gave the same response as above.)

Round 2

Reviewer 2 Report

Comments and Suggestions for Authors

The authors responsibly approached the revision and addition of the manuscript in accordance with my comments and recommendations, and gave the necessary explanations. The manuscript can be published in a journal.